# Systematic Review of the Apomorphine Challenge Test in the Assessment of Dopaminergic Activity in Schizophrenia

**DOI:** 10.3390/healthcare11101487

**Published:** 2023-05-19

**Authors:** Fabrice Duval

**Affiliations:** Pôle 8/9-APF2R, Centre Hospitalier, 68250 Rouffach, France; f.duval@ch-rouffach.fr; Tel.: +33-3-89-78-70-78

**Keywords:** schizophrenia, apomorphine test, prolactin, growth hormone, adrenocorticotropic hormone, cortisol

## Abstract

So far, neuroendocrine studies conducted in schizophrenic patients have yielded conflicting results. Many of these discrepancies may be explained by the diversity of factors that influence the hormonal levels (at baseline and in response to pharmacological stimuli), the heterogeneity of the populations studied, the absence of standardization of test challenges and the confounding and long-lasting effects of previous treatments. Numerous studies have used apomorphine (APO) in the evaluation of dopaminergic (DA) function in schizophrenic patients. APO, a direct acting DA receptor agonist, decreases prolactin (PRL) and stimulates growth hormone (GH), adrenocorticotropic hormone (ACTH) and cortisol secretion. Therefore, the magnitude of hormonal responses to APO is an indirect assessment of the functionality of DA receptors at the hypothalamic–pituitary level. This review provides an update on the applications of the APO test in schizophrenia in clinical, pathophysiological and therapeutic fields.

## 1. Introduction

The “original dopamine (DA) hypothesis” regarding the biological basis of schizophrenia (SCZ), formulated in the 1960s, postulated an enhancement of brain DA function [1]. A refinement of this hypothesis proposes that SCZ might be associated with an excess of subcortical DA activity (resulting in hyperstimulation of DA-D_2_ receptors and positive symptoms [including hallucinations and delusions]) and a deficit in cortical DA function (resulting in hypostimulation of DA-D_1_ receptors, negative symptoms [including anhedonia, lack of motivation, poverty of speech], and cognitive impairment) [2]. In other words, the “revised DA hypothesis” states that in SCZ, DA transmission is hypoactive in the prefrontal cortex and hyperactive in the mesolimbic areas. 

However, little is known about DA activity at the hypothalamic–pituitary level in SCZ. As illustrated in Figure 1, the hypothalamus—which together with the substantia nigra and the ventral tegmental area produces DA—is a region of the brain that receives its inputs from various areas (particularly the limbic system) and in turn releases both releasing and inhibiting hormones which act on the pituitary gland.

Apomorphine (APO), a non-selective short acting DA receptor agonist which stimulates D_1_-like (D_1_, D_5_) and D_2_-like (D_2S/2L_, D_3_, D_4_) receptors, has been used as a pharmacological probe of central DA function in psychiatric patients, especially in SCZs [3,4,5]. APO decreases prolactin (PRL) secretion (via D_2_ receptors of pituitary lactotrophs) and stimulates growth hormone (GH). APO—administered at a dose of 0.75 mg subcutaneously [4,6]—also stimulates adrenocorticotropic hormone (ACTH) and consequently cortisol secretion. Stimulation of GH and ACTH is secondary to agonization of D_1_-like and D_2_-like receptors on neurons in the hypothalamus which regulate the release of growth hormone-releasing hormone (GHRH) and corticotropin-releasing hormone (CRH) [4,7].

Despite tremendous advances in the investigation of the biology of schizophrenia made over the past four decades, the APO test by assessing the functionality of DA receptors at the hypothalamic–pituitary level—a region poorly explored by imaging techniques—still provides important clues to the biopathology underlying the processes involved in the heterogeneity of schizophrenia. Additionally, APO remains a useful tool for characterizing the effects of antipsychotic drugs on DA receptors. The aim of this review is therefore to provide an overview of the clinical and therapeutic applications of the APO challenge test in SCZ inpatients. 

## 2. Methods

### 2.1. Screening and Selection Process

A literature search was performed in the PubMed database for papers written in English and French prior to April 2023. The following search terms were used: (”prolactin” OR “growth hormone” OR “ACTH”OR “cortisol”) AND (“schizophrenia”) AND (“apomorphine”). 

### 2.2. Inclusion/Exclusion Criteria

Studies were included if participants were adults (≥18 years of age) with a diagnosis of schizophrenia. The exclusion criteria were: (1) non-original studies (including review articles); (2) no comparison condition; (3) preclinical studies; (4) no full-text available; (5) no drug-free period before performing the APO test. 

Figure 2 summarizes the selection process for studies included in the present review according to the Preferred Reporting Items for Systematic reviews and Meta-Analyses (PRISMA 2020) statement [8]. 

A total of 33 studies were eligible.

## 3. Literature Findings

### 3.1. Procedures

The effects of acute administration of APO in humans are known to be dose-dependent, including vomiting, hypothermia, yawning, penile erection, sedation, changes in motor activity and mood (see [7] for a review). Given poor oral bioavailability, owing the first-pass effect, APO is administered subcutaneously (SC) for use as a challenge test. In psychiatric patients, the rapid onset of action after SC injection, short half-life, and safety have led to the use of APO-induced hormonal responses to assess central DA activity. Despite the low doses used (≤0.75 mg SC), the most commonly reported side effects are nausea (and sometimes even vomiting), dizzying sensation, and a moderate drop in blood pressure (requiring blood pressure monitoring during the test). 

As mentioned in Table 1, the APO test is relatively well standardized since the dose administered in approximately 80% of studies is 0.75 mg SC (which is considered as a subemetic dose). Some studies, however, used 0.5 mg, which theoretically seems more appropriate for demonstrating hypersensitivity of DA receptors [9,10]. Usually, the APO test is performed in the morning (at 8 or 9 a.m.) in hospitalized patients, followed by blood samples collected during 80–160 min (at 15–30 min intervals) for assay of PRL and/or GH. In five studies [4,5,11,12,13], ACTH and/or cortisol were also measured (the minimum dose for ACTH/cortisol stimulation being 0.75 mg SC [6]). The maximum activity of APO on hormonal responses is observed between 15 and 30 min [4]. All studies cited in Table 1 evaluated unmedicated patients—the minimum drug-free period was 1 week, but in most studies the drug-free period was more than 2 weeks avoiding therefore the residual effects of previous medications (all patients included were not treated with long lasting antipsychotics). In this respect, normal baseline PRL level is a good index of the lack of residual impregnation of antipsychotics, given these drugs block D_2_ receptors and induce hyperprolactinemia. However, no study measured serum APO which could have ruled out pharmacokinetic differences as a source of bias in the results.

### 3.2. Hormonal Responses to Apomorphine in Schizophrenic Patients

Discrepant findings have been reported in the literature reflecting the heterogeneity of the populations studied, in terms of symptoms (positive vs. negative), duration and phase of the illness (acute vs. chronic), specific effect of stress or hospitalization, quality of antipsychotic withdrawal, and extraneous variables (age, gender, menstrual status, weight loss or gain).

#### 3.2.1. Prolactin Response

A decreased PRL suppression was reported by some investigators [15,24], particularly in chronic SCZs, but the vast majority of studies found no significant difference between SCZs and healthy controls (HCs) or patients with affective disorders [3,4,5,11,12,13,14,21,22,23,31]. The APO–PRL response does not predict clinical response to subsequent classical antipsychotic treatment [12]. However, higher PRL suppression at baseline has been reported in subsequent clozapine responders [32].

A major issue is that the magnitude of PRL suppression following APO is correlated with baseline, meaning that the higher the PRL baseline values, the greater the suppression [24]. This is why PRL suppression (PRLs) is frequently expressed as a percentage of change from baseline [3], rather than an area under the curve; by doing so, PRL values are no longer correlated with PRL baseline values (this also minimizes the gender effect since women often have higher baseline PRL levels [4]). 

#### 3.2.2. Growth Hormone Response

The GH response is usually expressed as GH peak value or difference between peak and baseline values [∆GH], more rarely as area under the curve. To avoid bias in the assessment of GH response, subjects with increased GH baseline (i.e., ≥4–6 mg/mL) are excluded from the analysis in most studies. 

Some studies [3,10,17,18,26], but not all [15,28], reported that acute SCZ were more prone to exhibit increased GH responses than chronic SCZs. Additionally, in a longitudinal study, it has been found an association between increased GH responses and relapse [20].

Chronic and sub-chronic SCZs, especially those with positive symptoms, frequently show reduced GH responses [4,5,11,17,23,28,31] compared to matched HCs, but this finding has not been unanimously reported [3,29]. In one study [3], GH peak was correlated with negative symptoms. 

Duration of the illness might also influence the APO–GH response since a negative correlation was found by Meltzer et al. [3]—however, this correlation disappeared when data were corrected for age.

Usually, GH response to APO decreases with age, both in patients (with various psychiatric diagnoses) and healthy subjects [11]. Consequently, age could be a confounding factor in the interpretation of the results. This could explain, in part, the increased GH responses reported in acute SCZs who are generally younger than chronic SCZs. 

Some studies suggest that the APO–GH response might be predictive of subsequent response to antipsychotic treatment. Regarding first-generation antipsychotics (i.e., neuroleptics [NLs]), it has been found that SCZs having high GH response subsequently failed to respond to NLs, while NLs responders had low GH response at baseline [32]; unfortunately, these results were not confirmed [12,17]. Lieberman et al. [33] reported that clozapine responders had, at baseline, greater GH response than non-responders. In one study, performed in patients with schizophrenic symptoms (i.e., SCZs or SADs), lithium responders had, at baseline, higher APO-induced GH stimulation than non-responders [34]; until now, this study has not been replicated. 

#### 3.2.3. Hypothalamic–Pituitary–Adrenal Hormone Response

So far, studies evaluating the ACTH and/or cortisol responses to APO consistently found reduced values in SCZs compared to HCs. The pioneering study of Mokrani et al. [4] showing decreased cortisol responses in SCZs was replicated by Meltzer’ s group [12], and all the studies we have performed subsequently also confirm this finding. The ACTH and cortisol responses are significantly reduced in SCZs with positive symptoms [4,5,11,13]. On the other hand, disorganized SCZs have comparable APO–ACTH/cortisol responses to those of HCs [4,13]. It can also be mentioned that unlike the PRL and GH responses, the ACTH/cortisol responses are neither influenced by the sex nor by the age of subjects. Moreover, the ACTH/cortisol responses appear independent of the hypothalamic–pituitary–adrenal (HPA) axis activity, as assessed by the serum cortisol response to the dexamethasone suppression test [4,5,11]. 

In one study [12], the APO-stimulated cortisol response was predictive of subsequent clinical response to antipsychotic treatment (either typical NLs or clozapine), responders having higher cortisol responses than non-responders.

### 3.3. Pathophysiology

As shown in Figure 1, APO has multiple sites of action at the hypothalamic–pituitary level: it inhibits directly PRL secretion from the pituitary and stimulates GH and ACTH/cortisol secretion via stimulation of GHRH and CRH from the hypothalamus. 

The blunted APO–PRL response found in some SCZs, which is unlikely to be due to a residual NL effect (given the normality of the baseline PRL value), could reflect a decreased functionality (or sensitivity) of D_2_ receptor of the lactotroph cells. This could be adaptive to increased activity of TIDA neurons (releasing an increased amount of presynaptic DA), possibly secondary to understimulation of D_2_ autoreceptors. Further studies are needed to confirm this hypothesis. However, it is also possible that PRLs blunting in some patients might be due to functional impairment of lactotrophs. Nevertheless, this hypothesis is not supported by the results of the protirelin (or TRH) tests, since the PRL responses to TRH tests (performed at 8 a.m. and 11 p.m.) are normal in SCZs [35].

Regarding the reduced APO–GH responses, often observed in chronic SCZs, it has been hypothesized a hyposensitivity of post-synaptic D_1_-like and D_2_-like receptors on neurons in the hypothalamus which regulate the release of GHRH [3,4,11]. Conversely, increased GH responses, more often observed in acute SCZs, would reflect a hypersensitivity of D_1_-like and D_2_-like receptors of GHRH neurons. However, the interpretation of these results is not unequivocal. D_1_-like and D_2_-like receptors are both stimulated by APO [36]. According to the “high output/low sensitivity” homeostatic principle [37], subsensitivity of D_1_-like and D_2_-like receptors would be adaptive of an excess of presynaptic DA release from the mesolimbic DA system and its hypothalamic innervation [4,5,11]. In other words, in the chronic phase of the disease, long-term DAergic hyperactivity could lead to desensitization of DA receptors. Given this blunting is more often observed in SCZs with positive symptoms, this is also consistent with the “revised DA hypothesis” stating that positive symptoms are associated with DA overactivity in the mesolimbic areas. How to interpret then the hypersensitivity of D_1_-like and D_2_-like receptors of GHRH neurons, often found in acute SCZ? A possible explanation would be that, in the acute phase of the disease, excessive DA release results in an overstimulation of the activity of DA receptors. Consistent with this hypothesis, it has been argued that the sensitivity of DA receptors depends on the entire DA neurons’ physiological and regulation mechanisms, including a double stimulation pattern, with a tonic and a phasic activity [38]. Some lines of evidence lead to consider that the tonic stimulation constitutes a background to sensitize the entire DA system to the phasic stimulation. Thus, if the tonic DA tone is increased, the sensitized neurons’ response to DA, or to DA agonists such as APO, is increased too. 

The APO–ACTH/cortisol response involves D_1_-like and D_2_-like receptors on neurons in the hypothalamus which regulate the release of CRH. Since ACTH and cortisol responses to APO are highly correlated [4,5,11,13], this suggests that cortisol stimulation by APO, despite localization of DA-D_2_ receptors in the adrenal gland, is secondary to that of ACTH. Moreover, it seems unlikely that reduced APO-induced ACTH/cortisol stimulation is due to decreased reserve of pituitary ACTH or residual antipsychotic effect. Thus, the blunted ACTH/cortisol response often observed in paranoid SCZs, but not in disorganized SCZs, may be related to functional impairment of post-synaptic D_1_-like and/or D_2_-like receptors secondary to enhanced synaptic DA transmission at the hypothalamic level. Although there is a relationship between ACTH/cortisol and GH responses, in both SCZs and HCs [11,36], the effect of APO on ACTH/cortisol involves different pathways from those of GH—in addition to their respective releasing hormones, other neurotransmitters/hormones such as acetylcholine, norepinephrine, serotonin, GABA, ghrelin and cholecystokinin are also involved to varying degrees in the GH response to APO [13]. 

### 3.4. Apomorphine as a Psychopharmacological Tool 

APO has a long history of use in preclinical and clinical drug studies [7]. The hormonal responses to APO make it possible to evaluate in vivo the mechanisms of action of different antipsychotic compounds on the hypothalamic–pituitary DA receptors (Figure 3).

NLs block D_2_ receptors leading to hyperprolactinemia [39] and a reduction in APO–PRL, GH [40,41] and ACTH/cortisol responses [41] (Figure 3a).Clozapine, which bind more “loosely” than DA to D_2_ receptors and dissociate more rapidly from D_2_ receptors than “thightly” bound antipsychotic drugs (i.e., NLs, risperidone) [42], slightly increases PRL [40,41], and has no significant effect on the APO–PRL, GH [41,43] and ACTH/cortisol responses [43] (Figure 3b).DA partial agonists, which bind tightly to D2 receptors but have lower intrinsic activity than full agonists [44], decrease PRL secretion via an intrinsic DA agonist action and block—like NLs, APO–PRL, GH and ACTH/cortisol responses [41]—via occupancy of D_2_ receptors (Figure 3c). Moreover, when aripiprazole (a high-affinity D_2_ receptor partial agonist [45]) is added (fixed dose 10 mg/d during 2 weeks) to ongoing antipsychotic therapy (with NLs or risperidone) in SCZs, PRL levels drop to normal levels, while the APO–PRL, GH, and ACTH/cortisol responses are not modified significantly [46] (Figure 3d). These latter results suggest that antipsychotics could be partially displaced from hypothalamic–pituitary D_2_ receptors by aripiprazole [47].

## 4. Discussion

From this review of the literature on the use of the APO challenge test in schizophrenic patients one may retain the following:Administered subcutaneously, APO, a short-acting DA receptor agonist, indirectly assesses—via the magnitude of induced hormonal responses—hypothalamic–pituitary DA receptor activity, at a given moment in the evolution of the disease, which results from both etiopathogenic processes and compensatory homeostatic mechanisms.Used in a clinical context, the APO test makes it possible to characterize the sensitivity of the DA receptors of different subgroups of patients: e.g., acute SCZs exhibit frequently higher GH responses than chronic SCZs (suggesting DA receptors’ hypersensitivity at the hypothalamic level); paranoid SCZs often show decreased ACTH/cortisol responses than disorganized SCZs (suggesting hyposensitivity of DA receptors connected to the HPA axis). However, to be pathophysiologically valid, the assessment of DA function requires controlling for the bias of insufficient antipsychotic withdrawal [4]. As it is now rare to have the possibility of evaluating hospitalized SCZs naïve to psychotropic drugs, this is probably the major limiting factor for the use of the APO test in the clinical field—this issue is, however, common to all exploration techniques in psychiatry. Another important point is the impact that the dose of APO administered can have on the evaluation of DA receptors in SCZ. Using 0.75mg SC, APO stimulates not only GH but also ACTH and cortisol secretion. APO-induced ACTH/cortisol secretion is consistently blunted in SCZs—particularly in paranoid SCZs—suggesting hyposensitivity of D_1_/D_2_-like receptors that may be adaptive to increased presynaptic DA release from the DA mesolimbic projections at the hypothalamic level. This blunting cannot be demonstrated with 0.5 mg of APO, since 0.75 mg is the minimum dose that stimulates ACTH secretion. Conversely, 0.5 mg might be more appropriate for demonstrating DA receptor hypersensitivity (often reported in acute SCZs) since 0.75 mg overstimulates DA receptors compared to 0.5 mg.Used as a psychopharmacological tool, APO is a relevant agent in the “in vivo” evaluation of the mechanisms of action of antipsychotic compounds since distinct specific response profiles characterize antipsychotic drugs based on their activity at D_2_ receptors.Nevertheless, the predictive value of the APO test in the subsequent response to antipsychotic treatment remains controversial owing to the small number of studies on this topic having produced conflicting or unreplicated results. To date, no studies have examined the predictive value of APO responses to partial D_2_ agonist drugs, whereas the therapeutic response to these compounds is likely influenced by the basal functionality of central D_2_ receptors.

Future research should determine the predictive value of the APO challenge test in order to rationalize the choice of pharmacotherapy by taking into account not only the clinical features but also the “biological state” as it can influence the therapeutic response. Better characterizing this state, by complementary paraclinical investigations such as the APO test, is therefore a major challenge in the context of “personalized medicine” for schizophrenia. 

## Figures and Tables

**Figure 1 healthcare-11-01487-f001:**
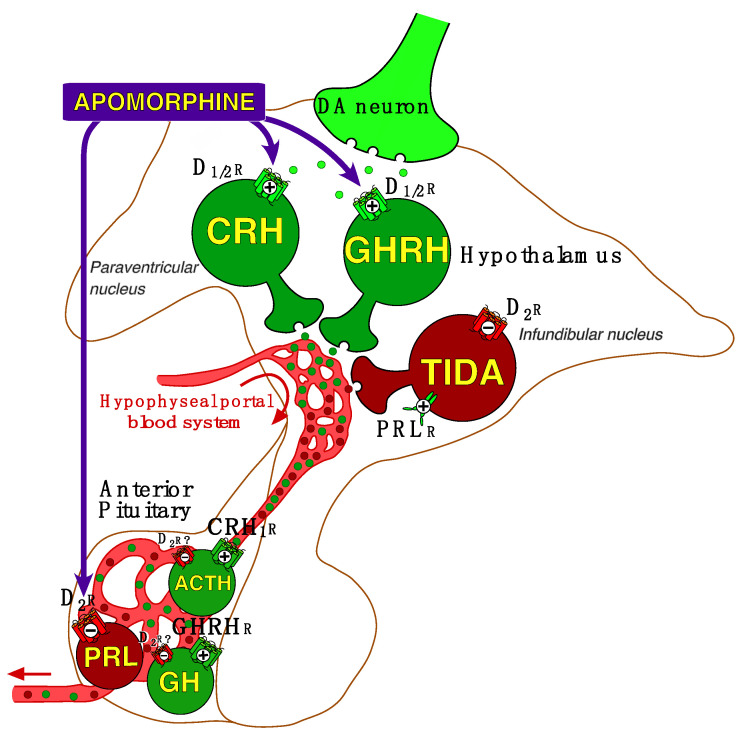
Overview of hypothalamic–pituitary regulation by dopamine (DA) and neuroendocrine targets of the apomorphine test. Apomorphine (administered subcutaneously) stimulates directly D_2_ receptors (D_2_R) of pituitary lactotrophs which inhibits the release of prolactin, and stimulates, via agonization of D_1_-like and D_2_-like receptors (D_1/2_R) at the hypothalamic level, growth hormone-releasing hormone (GHRH) and corticotropin-releasing hormone (CRH)—this latter stimulation requires a larger dose of apomorphine (≥0.75 mg) to be significant. Then, at the pituitary level, are stimulated growth hormone (GH), via GHRH receptors, and adrenocorticotropic hormone (ACTH), via CRH_1_ receptors. Additionally, ACTH stimulates the release of cortisol from the adrenal gland. The role of D_2_ receptors on GH and ACTH secretion at the pituitary level remains to be elucidated since they probably inhibit GH and ACTH release; this partly counteracts the stimulation operated by GHRH and CRH. On the other hand, PRL is tonically suppressed by DA originating from the tuberoinfundibular tract (TIDA). Via a short-loop feedback circuit, PRL stimulates TIDA neurons, resulting in an inhibition of its own secretion. An ultra-short feedback loop is provided by D_2_ autoreceptors that decrease the activity of TIDA neurons, maintaining stable homeostasis according to the ambient extracellular levels of DA.

**Figure 2 healthcare-11-01487-f002:**
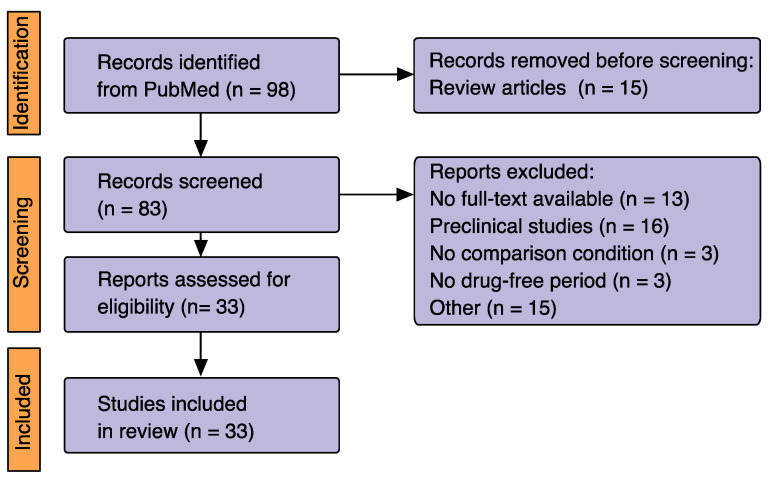
Flowchart of study selection process, adapted from the Preferred Reporting Items for Systematic Review and Meta-Analysis (PRISMA 2020) [8].

**Figure 3 healthcare-11-01487-f003:**
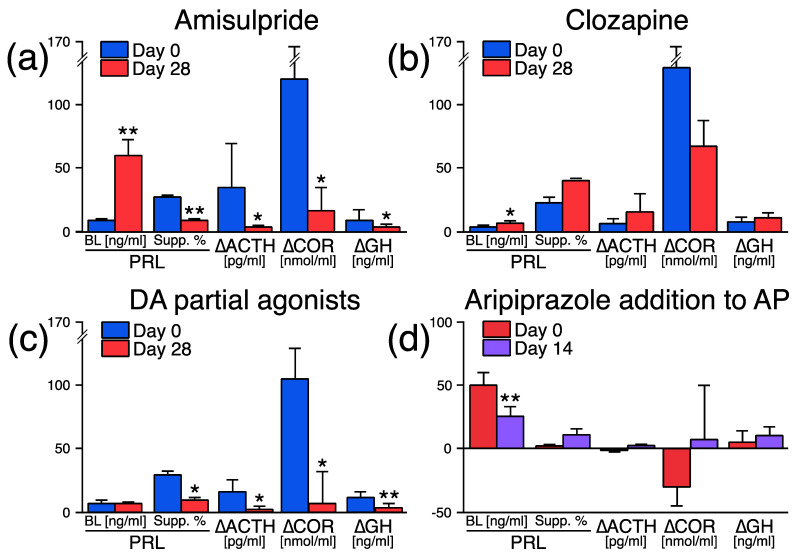
Effects of antipsychotics on the apomorphine (APO) test responses (0.75 mg SC). After a drug-free baseline period of at least 2 weeks (Day 0), schizophrenic patients (SCZs) received (**a**) amisulpride (n = 15, mean dose/day ± SD, 1247 ± 636 mg at day 28) [41]; or (**b**) clozapine (n = 7, 454 ± 10 mg) [43]; or (**c**) DA partial agonists: SDZ HDC-912 (n = 11, 4.0 ± 3.9 mg), OPC-4392 (n = 10, 17.2 ± 12.6 mg) [41]. In another experiment, aripiprazole (**d**) (10 mg/d) was added to ongoing antipsychotic (AP) treatment in 10 SCZs (see text) [46]. PRL, prolactin; Supp, % suppression from PRL baseline (BL) after APO injection; ∆ACTH, maximum adrenocorticotropin hormone increment from BL; ∆COR, maximum cortisol increment from BL; ∆GH maximum growth hormone increment from BL. Comparisons between baseline to endpoint, * *p* ≤ 0.05, ** *p* ≤ 0.01 (by Wilcoxon two-tailed signed rank test for paired samples).

**Table 1 healthcare-11-01487-t001:** Hormonal responses to apomorphine in schizophrenic patients.

Reference	Population Studied	APO Dose	Main Findings
Ettigi et al. [14] (1976)	17 chronic SCZs 21 HCs	0.75 mg SC	No difference in PRL suppression Lower GH stimulation in chronic SCZs
Rotrosen et al. [15] (1978)	22 chronic SCZs 9 HCs	0.5 mg SC	Blunted PRL suppression in chronic SCZs. Bimodal distribution of peak GH in SCZs
Pandey et al. [16] (1977)	9 acute SCZs 16 chronic SCZs 8 HCs	0.75 mg SC	Higher GH responses in acute SCZs than in HCs and chronic SCZs. Chronic SCZs had non-significant reduced GH responses than HCs.
Rotrosen et al. [17] (1979)	25 SCZs 16 HCs	0.5 mg SC	Blunted GH responses in chronic SCZs. Exaggerated GH responses in acute SCZs.
Meltzer et al. [18] (1982)	14 SCZs 8 SADs 16 HCs	0.75 mg SC	No difference in GH responses across groups.
Cleghorn et al. [19] (1983)	10 SCZs 14 HCs	Graded doses (0.1 to 0.75 mg SC)	Increased GH responses with lower doses of APO in SCZs with positive symptoms of psychosis.
Cleghorn et al. [20] (1983)	9 SCZs 9 HCs	0.75 mg SC	Exaggerated GH responses in SCZs prior to relapse.
Whalley et al. [21] (1984)	19 SCZs 9 HCs	0.75 mg SC	Higher PRL suppression in SCZs than in HCs. Greater GH responses in SCZs with Schneider’s first-rank symptoms than in those without and HCs.
Meltzer et al. [3] (1984)	40 SCZs 56 non-SCZs 16 HCs	0.75 mg SC	No difference in PRL suppression and GH stimulation across groups. Postive correlation between GH peak and negative symptoms in SCZs.
Müller-Spahn et al. [9] (1984)	11 chronic SCZs 11 HCs	0.5 mg SC	Higher GH response in HCs and SCZs after a 12-day drug-free period than in SCZs treated with NL.
Hitzemann et al. [22] (1984)	16 SCZs 13 SzDs	0.75 mg SC	No difference in PRL suppression.
Ferrier et al. [23] (1984)	30 SCZs 10 HCs	0.75 mg SC	Blunted GH responses in chronic SCZs. No difference in PRL suppression.
Davis et al. [24] (1985)	38 chronic SCZs 19 HCs	0.75 mg SC	Lower PRL suppression in SCZs than in HCs.
Zemlan et al. [25] (1986)	138 SCZs-SADs 10 HCs	0.75 mg SC	Higher GH responses in acute SCZs than in HCs and chronic SCZs.
Hirschowitz et al. [26] (1986)	71 SCZs-SADs-BDs 15 HCs	0.75 mg SC	Increased GH in SADs vs. BDs and SCZs.
Malas et al. [27] (1987)	16 SCZs 9 HCs	0.75 mg SC	Lower GH response in SCZs with poor premorbid psychosial functioning than HCs.
Brown et al [28] (1988)	12 SCZs 19 HCs	0.75 mg SC	No difference in GH responses. Lower GH response averaged over 12 trials in 5 SCZs vs. 5 HCs during longitudinal assessment (22 months).
Legros et al [29] (1992)	9 male SCZs 14 male HCs	0.5 mg SC	No difference in GH response.
Lieberman et al. [30] (1993)	70 acute SCZs 50 HCs	0.75 mg SC	Abnormal GH response in half of SCZs. Decreased GH response correlated with third ventricle enlargement.
Brambilla et al [31] (1994)	20 chronic and subchronic SCZs 9 HCs	0.5 mg SC	No difference in GH and PRL responses. However, blunted GH response. correlated with chronicity of the disorder. Negative symptoms correlated positively and positive symptoms correlated negatively with PRL responses.
Mokrani et al. [4] (1995)	46 SCZs 14 SADs 50 MDEs 18 HCs	0.75 mg SC	Lower ACTH and cortisol responses in SCZs and SADs vs. HCs and MDEs. Blunted GH responses (∆GH < 4 ng/mL) more frequent in SCZs than in MDEs. No difference in PRL suppression across groups, although BDs had lower values than MDEs.
Müller-Spahn et al. [10] (1998)	16 male acute SCZs 12 male HCs	3 dosages	Higher GH response in SCZs vs. HCs using 0.006 mg/kg SC APO, but not with 0.003 mg/kg and 0.012 mg/kg, suggesting increased DA receptor sensitivity in the hypothalamic–pituitary system in SCZs.
Duval et al. [11] (2000)	41 paranoid SCZs 74 MDEs 35 MDEPs 27 HCs	0.75 mg SC	Lower ACTH responses and higher rate of blunted GH responses in paranoid SCZs vs. HCs, MDEs, and MDEPs. No difference in PRL suppression across groups.
Meltzer et al. [12] (2001)	51–98 SCZs 15–25 HCs	0.01 mg/kg SC	Blunted cortisol response in SCZs Blunted GH, but not PRL, response in male SCZs.
Duval et al. [13] (2003)	20 SCZs 23 HCs	0.75 mg SC	Reduced ACTH and cortisol responses significantly more frequent in paranoid SCZs (n = 12) vs. disorganized SCZs (n = 8) and HCs. No difference in PRL suppression across groups.
Duval et al. [5] (2020)	13 male paranoid SCZs 13 male SADs 13 male BDs (depressed) 13 HCs	0.75 mg SC	Lower ACTH and cortisol stimulation in SCZs compared to HCs, but no difference in PRL suppression vs. other groups. There was a trend towards increased frequency of blunted GH response in SCZs.

APO, apomorphine; GH, growth hormone, PRL, prolactin, ACTH, adrenocorticotropin hormone; HCs, healthy control subjects; SCZs, schizophrenic patients; SzDs, patients with schizophreniform disorder; SADs, patients with schizoaffective disorder; BDs, patients with bipolar disorder; MDEs, patients with a major depressive episode; MDEPs, patients with a major depressive episode with psychotic features.

## Data Availability

Not applicable.

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
