# Peer review of "Systematic Review of the Apomorphine Challenge Test in the Assessment of Dopaminergic Activity in Schizophrenia"

_healthcare, 2023, doi:10.3390/healthcare11101487_

Round 1

Reviewer 1 Report

This manuscript is a systematic review of the apomorphine (APO) challenge test in the assessment of dopaminergic activity in schizophrenia. I have several questions.

1. In the part of introduction, the authors described as following, “APO—administered at a dose of 0.75 mg subcutaneously (SC)—also stimulates adrenocorticotropic hormone (ACTH) and consequently cortisol secretion. Stimulation of GH and ACTH is secondary to agonization of D1-like and D2-like receptors on neurons in the hypothalamus which regulate the release of growth hormone-releasing hormone (GHRH) and corticotropin-releasing hormone (CRH).” However, in the literatures, there were three doses of the APO (0.1mg SC, 0.5mg SC, 0.75mg SC) administration. Therefore, how about the potential clinical implications for the 0.75mg SC of APO administration on syndromes of schizophrenia?

2. How about the potential side effects or Serious Adverse Event for the APO treatment? Are there any modes of administration for APO, except for the subcutaneously administration?

3. Are there any meta-analyses for the randomized controlled trial (RCT) of APO treatment in schizophrenia?

4. Are there any matters needing attentions, considering the concomitant medication with APO treatment in the clinical practices?

5. How about the Animal or Human experiments indicated the changes of brain activity of dopamine (DA) before and after APO treatment?

Author Response

  • Comment #1:

            Indeed, different doses of apomorphine (APO) have been used in the evaluation of the dopamine (DA) system in schizophrenia (SCZ). As already stated in the manuscript, the 0.75 mg subcutaneously (SC) dose has been most often used. It is a subemetic dose which inhibits prolactin (PRL), and stimulates growth hormone (GH) and adrenocorticotropic hormone (ACTH) — and consequently cortisol — secretion. This last point is interesting since APO-induced ACTH/cortisol secretion is consistently blunted in SCZs, especially those with paranoid subtype, suggesting hyposensitivity of D1/D2-like receptors possibly adaptive to increased presynaptic DA turnover from mesolimbic DA projections at the hypothalamic level. This blunting cannot be evidenced using 0.5 mg of APO, given 0.75 mg is the minimum dose that stimulates ACTH secretion. Conversely, 0.5 mg could be more appropriate to demonstrate hypersensitivity of DA receptors (often reported in acute SCZs) since 0.75 mg overstimulates DA receptors compared to 0.5 mg.

Taking into account the comment of this referee, this aspect is now discussed more in depth in the “discussion section” (line 292): “Another important point is the impact that the dose of APO administered can have on the evaluation of DA receptors in SCZ. Using 0.75mg SC, APO stimulates not only GH but also ACTH and cortisol secretion. APO-induced ACTH/cortisol secretion is consistently blunted in SCZs—particularly in paranoid SCZs—suggesting hyposensitivity of D1/D2-like receptors that may be adaptive to increased presynaptic DA release from the DA mesolimbic projections at the hypothalamic level. This blunting cannot be demonstrated with 0.5 mg of APO, since 0.75 mg is the minimum dose that stimulates ACTH secretion. Conversely, 0.5 mg might be more appropriate for demonstrating DA receptor hypersensitivity (often reported in acute SCZs) since 0.75 mg overstimulates DA receptors compared to 0.5 mg.”

  • Comment #2:

         The purpose of this review was the use of APO as a challenge test in schizophrenia (i.e., acute use), and not the therapeutic effects of APO in SCZ (i.e., repeated use), in this context acute administration of low-dose of APO is generally well tolerated.

         Taking into account the comment of this referee, we added (first sentence of “procedures” paragraph, line 92): The effects of acute administration of APO in humans are known to be dose-dependent, including vomiting, hypothermia, yawning, penile erection, sedation, changes in motor activity and mood (see [7] for a review). Given poor oral bioavailability, owing the first-pass effect, APO is administered subcutaneously (SC) for use as a challenge test. In psychiatric patients, the rapid onset of action after SC injection, short half-life, and safety have led to the use of APO-induced hormonal responses to assess central DA activity. Despite the low-doses used (≤ 0.75 mg SC), the most commonly reported side effects are nausea (and sometimes even vomiting), dizzying sensation, and a moderate drop in blood pressure (requiring blood pressure monitoring during the test). ”

  • Comments #3 and following:

            The “antischizophrenic” effect of APO treatment has been reported by some but not all investigators (see Lal [7] for a review). However, these studies predate the 2000s and to our knowledge no meta-analysis for RCT of APO treatment in SCZ has been performed to date.

            Concerning the precautions to be taken regarding the combination with other drugs and the possible changes in brain activity during chronic treatment with APO, there are only few studies and they are in the field of neurology. In SCZ, APO has been found to increase prefrontal blood flow (Daniel et al., 1989) and increase cognitive activation of the anterior cingulate cortex (ACC) relative to HCs (Dolan et al., 1995)—ACC hypoactivity is commonly observed in SCZs and may be secondary to persistent illness and/or medication (Yücel et al., 2007).

Since these questions, related to administration of APO as a psychotropic treatment, are beyond the scope of this review, they will not be addressed in the revised version of the manuscript.

Reviewer 2 Report

Duval and Monreal have reviewed the evidence relating to the apomorphine challenge test in assessing dopaminergic activity in people with schizophrenia. The review is of interest and addressed an understudied area. In addition, the findings may have implications for other conditions where dopaminergic neurotransmission is involved.  I have several comments and suggestions for the authors. 

Major: 

1. For a systematic review the methods section is very short and I would suggest including more details. In addition, could the authors clarify if they have used the PRISMA guidelines for systematic reviews? I would also suggest including a figure showing manuscript included and excluded with the reason for this (or at least describe this in the methods).

2. Why were abbreviations for growth hormone and ACTH used as search terms?

3. Could the authors provide more details regarding the hypothesis and presumed usefulness of apomorphine challenge tests in people with schizophrenia. Is the aim e.g. to distinguish people with schizophrenia from other conditions or to use this a predictor for long-term outcomes? This could also be used to better structure the results. 

4. Lines 163/164 ''Moreover, the ACTH/cortisol responses appear independent of the hypothalamic-pituitary-adrenal (HPA) axis activity [4,5,28,29]''. Could the authors speculate on these findings? How do they interpret this and could this suggest an involvement of the suprachiasmatic nuclei which are known to influence ACTH sensitivity of the adrenal cortex? 

5. Overall, the review seems quite limited and would benefit from a more in-depth discussion on the possible mechanisms and usefulness of the apo challenge test in people with schizophrenia. 

Minor: 

1. Figure 1 'arcuate nucleus'. The preferred term in humans would be 'infundibular nucleus'. 

2. Where was figure 2 taken from? Was this reproduced from original research? If so, this should be referenced. 

Author Response

  • Comment #1:

            Taking into account the comment of this referee we now reformulated the following sentence (line 78): “the exclusion criteria were: (1) non-original studies (including review articles); (2) no comparison condition; (3) preclinical studies; (4) no full-text available; (5) no drug-free period before performing the APO test”.

We added in the “methods section” a figure (Figure 2) which details the included and excluded studies. This figure is adapted from the PRISMA 2020 flowchart, and the following sentence was added (line : “Figure 2 summarizes the selection process for studies included in the present review according to the Preferred Reporting Items for Systematic reviews and Meta-Analyses (PRISMA 2020) statement [8]”.

We also added the following reference: [8] Page, M.J.; McKenzie, J.E.; Bossuyt, P.M.; Boutron, I.; Hoffmann, T.C.; Mulrow, C.D.; Shamseer, L.; Tetzlaff, J.M.; Akl, E.A.; Brennan, S.E.; Chou, R.; Glanville, J.; Grimshaw, J.M.; Hróbjartsson, A.; Lalu, M.M.; Li, T.; Loder, E.W.; Mayo-Wilson, E.; McDonald, S.; McGuinness, L.A.; Stewart, A.; Thomas, J.; Tricco, A.C.; Welch, V.A.; Whiting, P.; Moher, D. The PRISMA 2020 statement: An updated guideline for reporting systematic reviews. J. Clin. Epidemiol. 2021,134,178-189. DOI: 10.1016/j.jclinepi.2021.03.001.

   The references have therefore been renumbered.

  • Comment #2:

In fact, the use of the term "growth hormone" or "GH", or "adrenocorticotropic hormone" or "ACTH" did not change the PubMed results. However, in the revised version, following the comment of this referee, we corrected "GH" to "growth hormone" in the search terms, but left the term "ACTH" (since until now we are the only research group to measure ACTH following administration of APO in psychiatric patients).

  • Comment#3

            The aim of this review was to provide an overview of the clinical and therapeutic applications of the APO challenge test in SCZ inpatients.  As already stated in the manuscript, the usefulness of the APO test in psychiatric patients is to assess the central DA receptor function. APO test results cannot make a psychiatric diagnostic (in our current state of knowledge, developing a “biological symptomatology” is out of the question), but may clarify hypothalamic-pituitary DA receptor functionality in patients considered as a group (e.g., SCZ vs. HC) or subgroups (acute vs. chronic; paranoid vs. disorganized). Indeed, biology cannot substitute for clinical observation, and is only meaningful when interpreted in a clinical context. On the other hand, studies, using the APO test as a predictor of long-term outcomes are scarce, but it could be a promising research topic. This review suggests the use of the APO test as a predictor especially to partial DA agonists (as already stated in the discussion section), but this is not the main topic of this review.

Taking into account the comment of this referee, we modified the last sentence of the manuscript: “Future research should determine the predictive value of the APO challenge test in order to rationalize the choice of pharmacotherapy by taking into account not only the clinical features but also the “biological state” as it can influence the therapeutic response. Better characterizing this state, by complementary paraclinical investigations such as the APO test, is therefore a major challenge in the context of “personalized medicine” for schizophrenia”.

  • Comment #4:

            This is not a speculation since we have previously studied the interactions between the HPA axis activity and the hormonal responses to APO in psychiatric patients (e.g., SCZs, MDDs, depressed bipolar patients, SADs) and HCs.

We added in the revised version the following sentence “Moreover, the ACTH/cortisol responses appear independent of the hypothalamic-pituitary-adrenal (HPA) axis activity, as assessed by the serum cortisol response to the dexamethasone suppression test [4,5,29,30]”. 

  • Comment #5:

            In our opinion, this present review (revised) lists the various applications of the APO challenge test in schizophrenic patients in a concise and easy-to-understand manner even for non-specialists.

  • Comment #6: 

    Arcuate nucleus has been replaced by infundibular nucleus as suggested by this referee.

    Figure 2 synthesizes different studies performed by our group. “Ad hoc” references are now mentioned.

Round 2

Reviewer 1 Report

None.

Reviewer 2 Report

I would like to thank the authors for taking the time to reply to the comments and suggestions raised and for updating their manuscript. I have no further suggestions.